# Prospective and Cross-Sectional Factors Predicting Caregiver Motivation to Vaccinate Children with Attention-Deficit/Hyperactivity Disorder against COVID-19: A Follow-Up Study

**DOI:** 10.3390/vaccines12050450

**Published:** 2024-04-23

**Authors:** Tai-Ling Liu, Ray C. Hsiao, Wen-Jiun Chou, Cheng-Fang Yen

**Affiliations:** 1Department of Psychiatry, Kaohsiung Medical University Hospital, Kaohsiung Medical University, Kaohsiung 80754, Taiwan; 2Department of Psychiatry, School of Medicine, College of Medicine, Kaohsiung Medical University, Kaohsiung 80708, Taiwan; 3Department of Psychiatry, Seattle Children’s, Seattle, WA 98195, USA; 4Department of Psychiatry and Behavioral Sciences, School of Medicine, University of Washington, Seattle, WA 98105, USA; 5Department of Child and Adolescent Psychiatry, Chang Gung Memorial Hospital, Kaohsiung Medical Center, Kaohsiung 83341, Taiwan; 6School of Medicine, Chang Gung University, Taoyuan 33302, Taiwan; 7College of Professional Studies, National Pingtung University of Science and Technology, Pingtung 91201, Taiwan

**Keywords:** attention-deficit/hyperactivity disorder, vaccine, COVID-19

## Abstract

Adolescents with attention-deficit/hyperactivity disorder (ADHD) have higher risks of contracting COVID-19 and worse outcomes compared with adolescents without ADHD. The most effective method of preventing infection is vaccination. This follow-up study explored the prospective and cross-sectional factors influencing caregiver willingness to vaccinate children with ADHD against COVID-19. Baseline data on caregiver demographics, affiliate stigma, parenting stress, emotional difficulties, beliefs regarding the causes of ADHD, and ADHD symptoms were collected prior to the outbreak of the COVID-19 pandemic in Taiwan. At follow-up, the study assessed caregiver willingness to vaccinate children with ADHD, the challenges caregivers faced in parenting during the pandemic, and ADHD symptoms. The results revealed that caregiver age at baseline was positively associated with a willingness to vaccinate children against COVID-19 at follow-up. By contrast, the belief that ADHD resulted from failures in parental discipline at baseline was negatively associated with caregiver willingness to vaccinate. Parenting challenges were also negatively associated with caregiver willingness to vaccinate. Therefore, the age of caregivers, beliefs about the causes of ADHD, and parenting challenges during the pandemic should be considered when developing interventions to enhance caregiver willingness to vaccinate children with ADHD.

## 1. Introduction

COVID-19 emerged in late 2019 and devastated populations globally. The World Health Organization declared on 3 May 2023, that COVID-19 was no longer a public health emergency of international concern. A total of 774,834,251 confirmed cases of COVID-19 and 7,037,007 COVID-19-associated deaths worldwide had been reported as of 15 March 2024 [1], and these numbers have continued to increase [2]. Older individuals have been identified as the population with high mortality and morbidity rates of COVID-19 [3]. In addition to older individuals, individuals with chronic diseases, such as hypertension, chronic obstructive pulmonary disease, obesity, cerebrovascular disease, chronic liver disease, chronic renal disease, human immunodeficiency virus infection, tuberculosis and cancer, have a greater probability of mortality and morbidity after contracting COVID-19 [4,5,6]. Compared to adults, children have a lower rate of contracting COVID-19; however, children remain susceptible to COVID-19 [7,8], especially those with comorbidities of chronic physical and mental illnesses [9]. The pandemic affected their physical health [7], mental health [10,11], educational experiences [12], and family dynamics [13]. Caregivers played a crucial role in taking care of children and adolescents during the COVID-19 pandemic.

One caregiver responsibility is deciding on vaccinating children against COVID-19. Vaccination is a crucial strategy to mitigate the risk of contracting COVID-19 and related hospitalizations [14,15,16]. The World Health Organization advises vaccinating children over 6 months of age against COVID-19 [14]. Although most caregivers are willing to vaccinate their children to decrease the COVID-19 infection risk, many remain undecided or hesitant about vaccination [17,18,19,20,21,22,23,24,25]. A meta-analysis revealed that 60.1% of caregivers intended to vaccinate their children against COVID-19, with 22.9% having no such intention and 25.8% uncertain [26]. The Theory of Planned Behavior (TPB) identifies several factors that influence caregiver motivation to vaccinate against COVID-19, comprising attitudes toward the vaccination of children, perceived judgment from other family members and friends regarding vaccination, and confidence in a vaccine’s efficacy [27]. Additionally, the Protection Motivation Theory suggests that both threat appraisal, such as caregiver concerns about COVID-19 infection, and coping appraisal, such as confidence in vaccine efficacy, affect caregiver motivation to vaccinate [28,29,30]. Consequently, studies have demonstrated that caregivers with heightened perceptions of the threat of COVID-19, favorable vaccination [26], attitudes, minimal vaccine side effect fears, a strong desire to protect their children [31], and positive societal vaccination views are more inclined to vaccinate their children. Furthermore, non-TPB factors, including being a father, being an older caregiver, and having a higher income, have also been linked to increased motivation to vaccinate children against COVID-19 [26]. Hence, factors outside the direct influence of the COVID-19 pandemic that may affect caregiver vaccination motivation must be further investigated.

Studies have revealed that children and adolescents with attention-deficit/hyperactivity disorder (ADHD) are at increased risk of contracting COVID-19 and experiencing adverse outcomes, including a higher severity of symptoms and a greater likelihood of hospitalization, than those without ADHD [32,33,34]. These results suggest that symptoms of inattention contribute to the challenges faced by children and adolescents with ADHD in consistently wearing face masks and maintaining social distancing to prevent COVID-19 infection [32,35]. Despite the necessity for children and adolescents with ADHD to receive COVID-19 vaccinations to reduce infection risk, approximately one-fourth of caregivers of children with ADHD exhibited hesitancy toward vaccination, and more than 10% were opposed to vaccinating their children against COVID-19 [24]. Moreover, concerns regarding the safety of COVID-19 vaccines among caregivers [24], in addition to irregular adherence to ADHD medication and comorbid oppositional defiant disorder (ODD) symptoms in adolescents under their care, are cross-sectionally associated with caregiver willingness to vaccinate children against COVID-19 [24].

Several factors influencing caregiver motivation to vaccinate their children against COVID-19 warrant further investigation. First, no study to date has prospectively examined the effects of prepandemic factors on caregiver motivation to vaccinate adolescents with ADHD against COVID-19. Second, the dynamics between caregivers’ affiliate stigma, parenting stress, and emotional challenges (such as depression and anxiety) prior to the COVID-19 pandemic and their motivation to vaccinate adolescents with ADHD have not been explored. Caregivers of children and adolescents with ADHD frequently encounter public stigma and may internalize this negative perception, resulting in affiliate stigma [36]. Affiliate stigma can exacerbate caregiver parenting stress [37] and reluctance toward the diagnosis and treatment of ADHD for their children [38]. Affiliate stigma may also erode caregivers’ confidence, reducing their motivation to vaccinate their children against COVID-19. Moreover, the Multimodal Treatment Study of Children with ADHD indicated that mothers of children with ADHD experienced higher levels of parenting stress than did mothers of children without ADHD [39]. Increased parenting stress may weaken caregiver capacity and willingness to care for children, including vaccination against COVID-19. Depression and anxiety are also prevalent among caregivers of children and adolescents with ADHD [40], particularly during times of increased stress, such as the COVID-19 pandemic [41]. These emotional problems may demotivate caregivers, undermining their confidence and ability to provide necessary care, including vaccinations. However, the hypothesis that prepandemic affiliate stigma, parenting stress, and emotional problems predict caregivers’ motivation to vaccinate their adolescents with ADHD against COVID-19 requires further examination.

Third, studies have indicated that although caregivers most commonly attribute ADHD to brain dysfunction, they also consider nonbiological factors such as inadequate parental discipline and the education system’s focus on academic achievement as causes [42,43]. Caregiver perceptions of the origins of children’s behavioral challenges affect their willingness to pursue interventions [44]. However, whether the attribution of ADHD’s etiologies by caregivers influences their motivation to vaccinate children against COVID-19 is poorly understood. Fourth, during the COVID-19 pandemic, caregivers of children with ADHD faced multifaceted stressors, which may have prevented them from recognizing the importance of vaccination. This hypothesis warrants further investigation.

The Study on the Mental Health of Caregivers of Adolescents with ADHD, conducted before the outbreak of COVID-19 in Taiwan, collected data on affiliate stigma, parenting stress, emotional difficulties, and the perceived causes of ADHD in caregivers of adolescents with ADHD [37,43]. The present study used those data to assess the predictive value of prepandemic factors (i.e., caregiver demographics, affiliate stigma, parenting stress, emotional difficulties, and perceived ADHD causes) and ADHD and ODD in adolescents on caregiver motivation to vaccinate their children against COVID-19. The study also examined the associations between caregiver parenting challenges during the pandemic, emotional difficulties, ADHD and ODD in adolescent children, and medication adherence at follow-up with caregiver motivation to vaccinate their children. The authors hypothesize that prepandemic factors such as affiliate stigma, parenting stress, emotional difficulties, and caregiver perceptions of ADHD causes, in addition to ADHD and ODD symptoms in adolescents, predict caregiver willingness to vaccinate their children against COVID-19. Furthermore, the authors hypothesized that the current challenges related to the pandemic faced by caregivers, including their emotional difficulties and the behavioral symptoms and medication adherence of their adolescent children with ADHD, were associated with caregiver motivation to vaccinate.

## 2. Methods

### 2.1. Participants

In the Study on the Mental Health of Caregivers of Adolescents with ADHD, participants were recruited from three child and adolescent psychiatric outpatient clinics in two general hospitals in southern Taiwan between June 2018 and May 2021. Individuals were included if they were the primary caregiver of an adolescent aged between 10 and 16 years with ADHD according to the *Diagnostic and Statistical Manual of Mental Disorders, Fifth Edition (DSM-5)* [45] and the medical decision makers for this adolescent. Individuals were excluded if they had a schizophrenia spectrum disorder, intellectual disability, substance use disorder, or other physical or psychiatric condition that could impair their comprehension of the study goals and methodology or if they were a caregiver of an adolescent with a comorbid intellectual disability, autism spectrum disorder, bipolar spectrum disorder, or schizophrenia spectrum or other psychotic disorder.

A total of 194 caregivers of adolescents with ADHD participated in the initial assessment. Before May 2021, Taiwan experienced a relatively mild COVID-19 outbreak, with only 0.037% of the population infected [46], allowing the baseline data to reflect the circumstances of the majority of Taiwanese caregivers of adolescents with ADHD before the COVID-19 pandemic. Subsequently, the same 194 caregivers were invited for a follow-up interview conducted between June 2023 and February 2024. Trained research assistants briefed the participants on the objectives and procedures of the follow-up study. Participants were provided with self-reported questionnaires, received instructions for completion, and filled out the questionnaires in individual research rooms.

### 2.2. Ethics Statement

The present study was approved by the Institutional Review Boards of Chang Gung Medical Foundation (protocol number, 202201110B0; date of approval, 2 August 2022) and Kaohsiung Medical University Hospital (protocol number, KMUHIRB-E(I)-20210342; date of approval, 10 February 2022). Participants provided written informed consent for their involvement. The questionnaire-based study did not involve experiments on humans or human tissue samples. The study adhered to the principles of the Declaration of Helsinki and the guidelines for the Conduct, Reporting, Editing, and Publication of Scholarly Work in Medical Journals.

### 2.3. Measures

#### 2.3.1. Outcome Variable: Caregiver Motivation to Have Their Children Vaccinated against COVID-19

The present study employed the nine-item parent version of the Motors of COVID-19 Vaccination Acceptance Scale (P-MoVac-COVID19S) [47] to assess caregiver motivation to vaccinate children against COVID-19. Participants rated each item on a seven-point scale with endpoints ranging from 1 (strongly disagree) to 7 (strongly agree). A cumulative higher score suggests increased caregiver motivation to have their children vaccinated against COVID-19. The P-MoVac-COVID19S had a Cronbach’s α value of 0.91.

#### 2.3.2. Predicting Variables at Baseline

##### Affiliate Stigma Scale (ASS)

The 22-item ASS is a self-administered questionnaire that assesses caregiver internalized stigma associated with a family member’s mental health condition [30]. This study used the ASS to assess caregivers’ affiliate stigma regarding their children’s ADHD. Agreement with each item on the ASS was rated using a four-point scale with endpoints ranging from 1 (*strongly disagree*) to 4 (*strongly agree*). A cumulative higher score suggests a greater caregiver affiliate stigma. The original iteration of the ASS exhibited excellent internal consistency (Cronbach’s α value: 0.94) and satisfactory predictive validity [36]. The Cronbach’s α value for the ASS in this study was 0.91.

##### Parenting Stress Index, Fourth Edition Short Form

This study used the Traditional Chinese version [48] of the Parenting Stress Index, Fourth Edition Short Form (PSI-4-SF) [49] to evaluate the levels of caregiver-reported parenting stress in three domains, including parental distress (caregiver distress due to their personal factors), dysfunctional interactions between caregivers and their children (caregiver dissatisfaction with interactions with their children and attitudes toward their children’s unacceptable behaviors), and difficult children (caregivers’ perceived children’s difficulties in self-regulation). Each item was rated on a five-point scale with endpoints ranging from 1 (*strongly disagree*) to 5 (*strongly agree*). Higher total subscale scores indicate increased parenting stress. The original PSI-4-SF had acceptable to satisfactory internal consistency (Cronbach’s α of three subscales: 0.88–0.90) and acceptable to satisfactory test–retest reliability (test–retest coefficient of three subscales: 0.68–0.85) [49]. Similarly, the Traditional Chinese version exhibited acceptable to satisfactory internal consistency (Cronbach’s α of three subscales: 0.86–0.91) [48]. The Cronbach’s α values for the three subscales ranged from 0.82 to 0.88. The present study transformed the scores of the three PSI-4-SF subscales into a score representing parenting stress using factor analysis.

##### Beck Depression Inventory-II (BDI-II)

The present study used the 21-item BDI-II to assess caregivers’ severity of depressive symptoms over the 2 weeks before assessment [50]. A cumulative higher score indicates more severe depression. The Cronbach’s α for the BDI-II in this study was 0.86.

##### Beck Anxiety Inventory

The 21-item Beck Anxiety Inventory (BAI) was used to assess symptoms of anxiety over the 2-week period preceding the study [51]. A higher total BAI score indicates more severe anxiety. The Cronbach’s α for the BAI was 0.88.

##### Caregiver-Attributed Etiologies of ADHD

The study explored whether participants believed that brain dysfunction, the failure of caregivers in disciplining the children, or the education system’s overemphasis on academic performance (credentialism) were reasons for adolescents to be diagnosed with ADHD [42]. Participants could respond to this question with “yes” or “no.”

##### Traditional Chinese Version of Swanson, Nolan, and Pelham, Version IV Scale (TC-SNAP-IV)

The present study used the 26-item TC-SNAP-IV to evaluate the severity of DSM-5–derived ADHD and ODD symptoms reported by caregivers for the children in their care over the preceding month [52,53]. Caregivers rated each item on a four-point scale with endpoints ranging from 0 (*symptom absent*) to 3 (*symptom severe*). The Cronbach’s α values for the subscales of inattention, hyperactivity and impulsivity, and ODD symptoms were 0.87, 0.88, and 0.90, respectively.

##### Caregiver and Adolescent Demographics

Information regarding the sex and age of the caregivers, the sex and age of the adolescents in their care, and caregiver educational level (high school or below vs. college or above) was collected.

#### 2.3.3. Predicting Variables at Follow-Up

Depression was reassessed using the BDI-II, anxiety was reassessed using the BAI, and the severity of ADHD and ODD symptoms was reassessed using the SNAP-IV. The study measured participants’ parenting challenges during the COVID-19 pandemic using the 31-Item Parenting Difficulties in Infectious Disease Pandemic Inventory (PDIDPI) [54]. The PDIDPI encompasses seven subscales: infection, school and learning, life changes, care burden, daily activities, health care access, and emotional and behavioral responses [54]. Items were rated on a four-point scale with endpoints ranging from 0 (*not challenging at all*) to 3 (*extremely challenging*). A high total subscale score indicates increased parenting challenges during the COVID-19 pandemic. The scores from the seven PDIDPI subscales were transformed into an overall score indicating parenting difficulties during the COVID-19 crisis using factor analysis. Additionally, the study solicited participants to assess adolescents’ ADHD medication adherence, with adherence rated on a four-point scale with endpoints ranging from 1 (*never adhering*) to 4 (*consistently adhering*).

### 2.4. Statistical Analysis

Statistical analyses were performed using IBM SPSS Statistics version 24.0 (IBM Corporation, Armonk, NY, USA). The demographic characteristics, caregiver motivation to vaccinate children against COVID-19, affiliate stigma, parenting stress, depression, anxiety, attributed causes of ADHD, challenges in parenting during the pandemic, and adolescents’ ADHD and ODD symptoms and medical adherence are presented as means (standard deviations) and frequencies (percentages). To determine the normal distribution of continuous variables, criteria of absolute values of <7 and <3 for kurtosis and skewness, respectively, were applied [55]. These tests did not reveal significant deviations. The present study used bivariable linear regression analysis to examine the prospective and cross-sectional associations between independent factors and caregiver motivation to vaccinate children against COVID-19. Factors significantly associated with this motivation were further analyzed in a multivariable linear regression model. A two-tailed *p* value of <0.05 was considered statistically significant.

## 3. Results

In total, 98 participants (75 women and 23 men), representing 50.5% of the initial group, participated in the follow-up study. In addition, 10 (5.2%) refused participation, and 86 (44.3%) stopped visiting the outpatient clinics and were lost to follow-up. The study observed no differences at baseline between those who did and did not complete the follow-up survey in terms of caregivers’ sex (χ^2^ = 0.783, *p* = 0.376), age (t = 0.984, *p* = 0.326), level of education (χ^2^ = 2.517, *p* = 0.113), affiliate stigma (t = 0.068, *p* = 0.945), parenting stress (t = −0.754, *p* = 0.451), depression (t = 0.437, *p* = 0.663), anxiety (t = 1.085, *p* = 0.279), and beliefs regarding the causes of ADHD (χ^2^ = 0.005~0.687, *p* = 0.407~0.943). Similarly, no baseline differences were observed in the adolescents’ sex (χ^2^ = 1.822, *p* = 0.177), inattention levels (t = 0.074, *p* = 0.941), hyperactivity/impulsivity (t = −0.525, *p* = 0.600), and symptoms of ODD (t = 1.126, *p* = 0.262) between the groups that did and did not participate in the follow-up. However, caregivers who did not participate in the follow-up were more likely to have older children with ADHD (t = 2.807, *p* = 0.006).

Table 1 presents the caregiver and adolescent demographics, caregiver experiences of affiliate stigma, parenting stress, depression, anxiety, perceptions regarding the etiology of ADHD, challenges faced during the pandemic, motivation to vaccinate children, and the ADHD and ODD symptoms of adolescents at baseline and follow-up. The average motivation score among caregivers for vaccinating their children against COVID-19 was 50.6 (standard deviation = 10.1), with scores ranging from 15 to 63.

Table 2 presents the results of the prospective analysis of factors and cross-sectional prediction of caregiver motivation to vaccinate their children against COVID-19. The analysis revealed that caregiver age at the start of the study was positively correlated with caregiver motivation to vaccinate children against COVID-19 at the follow-up stage (*p* = 0.048). By contrast, caregivers attributing ADHD’s etiology to disciplinary failures at baseline had less motivation to vaccinate at follow-up compared with those without this belief (*p* = 0.028). Other factors, such as those related to the caregiver (sex, educational level, affiliate stigma, parenting stress, depression, and anxiety) and the adolescent (sex, age, and ADHD or ODD symptoms at baseline), were not significantly associated with the motivation to vaccinate (all *p* > 0.05). Both caregiver age and the attribution of the etiology of ADHD to disciplinary failure were included in a multivariable regression analysis. This analysis confirmed that both caregiver age (*p* = 0.049) and the attribution of ADHD to disciplinary failure (*p* = 0.029) were significant predictors of the motivation to vaccinate children against COVID-19.

Regarding the cross-sectional factors assessed at follow-up, parenting difficulties encountered during the COVID-19 pandemic were negatively associated with the motivation to vaccinate (*p* = 0.022). Other factors were not significantly associated with the motivation to vaccinate (*p* > 0.05).

## 4. Discussion

The present study demonstrated that caregivers of adolescents with ADHD who were older at baseline had increased motivation to vaccinate children against COVID-19 at follow-up. By contrast, caregivers who attributed the etiology of ADHD to disciplinary failures at baseline demonstrated reduced motivation to vaccinate at follow-up. Additionally, substantial parenting challenges during the COVID-19 pandemic were associated with diminished willingness to vaccinate.

The results of the analysis indicate that older caregivers were more willing to vaccinate their children against COVID-19. This observation is consistent with the conclusions of a systematic review and meta-analysis that included 44 studies and 317,055 caregivers [26]. Given the elevated risk of severe COVID-19 symptoms requiring intensive care or leading to death among older individuals, older individuals were a focal point for vaccination advocacy [56,57]. Moreover, one review indicated that the younger age group (≤40 years) experienced more psychological distress during the pandemic [58]. Young caregivers lack the life experience to manage the multiple effects of the COVID-19 pandemic. These difficulties reduced young caregivers’ ability to obtain vaccination information for adolescents and make informed vaccination decisions. Although some studies have suggested that mothers exhibited more vaccine hesitancy due to concerns over potential vaccine side effects in children [26,59,60], the present study did not identify caregiver sex as a factor influencing vaccination motivation.

The present study also highlighted a lower motivation to vaccinate children against COVID-19 at follow-up in caregivers who believed ADHD to be the result of disciplinary failures than in those who did not share this belief. This insight is consistent with Attribution Theory, which posits that experience strongly influences caregiver decisions [61]. Caregivers who believe ADHD to be a result of disciplinary failures may feel discouraged and lack confidence in their ability to discipline or vaccinate their children. This lack of confidence may further reduce their interactions with teachers and parents of other children, limiting their access to information and norms about COVID-19 vaccination for children.

The present study discovered that high parenting difficulties during the COVID-19 pandemic were cross-sectionally associated with reduced motivation to vaccinate children against COVID-19 in caregivers. At least one study suggested that caregivers of children and adolescents with ADHD faced numerous parenting challenges during the COVID-19 pandemic [54]. Challenges such as managing children’s emotions and behaviors, conflicts within the family, adhering to COVID-19 protective measures, supporting learning and daily activities, and regulating Internet use heightened the risk of mental health problems among caregivers, and substantial time and effort were required to manage these challenges effectively [41]. Consequently, these challenges may have diminished caregiver willingness to vaccinate their children. By contrast, a lack of willingness to vaccinate may have heightened the risk of COVID-19 infection for adolescents, caregivers, and other family members, exacerbating parenting difficulties. Caregiver personality traits and insufficient social support were also strongly associated with reduced willingness to vaccinate and increased parenting challenges during the pandemic.

The present study hypothesized that caregiver prepandemic affiliate stigma, parenting stress, and emotional problems, in addition to the ADHD and ODD symptoms of adolescents under their care, predicted caregiver motivation to vaccinate children with ADHD against COVID-19. However, the results did not support these hypotheses. One possible explanation is the variability of these factors, reducing their predictive power regarding vaccination motivation over time. Moreover, this study observed no cross-sectional associations between caregivers’ emotional problems, adolescents’ ADHD and ODD symptoms, and vaccination motivation, suggesting that further investigation is required to understand the effects of these factors on vaccination decisions.

This study is the first to investigate the predictive power of various factors on caregiver willingness to vaccinate children with ADHD against COVID-19. Based on the results of this study, the authors propose the following recommendations. Health professionals must formulate strategies to enhance willingness among caregivers to vaccinate their children with ADHD against COVID-19. Priority should be placed on intervention programs targeting younger caregivers, those who attribute ADHD to disciplinary failures, and those who experience parenting challenges in managing their children’s behavior during the pandemic. Health care professionals must understand caregivers’ perceived etiologies of ADHD and assist them in shaping attributions that facilitate access to treatment. Moreover, these findings underscore the requirement for programs designed to enhance parenting skills to alleviate stress during the pandemic.

This study has several limitations. First, the data collection relied solely on self-reports from caregivers of children with ADHD, potentially introducing single informant bias and social desirability bias. Future studies can enhance validity by incorporating objective measures of vaccination status and corroborating self-reported data with medical records. Additionally, employing qualitative methods, such as interviews or focus groups, could provide deeper insights into the motivations behind caregiver attitudes and decisions. Second, although this prospective study examined the associations between baseline factors and subsequent vaccination motivation, it did not establish causality. Unexamined variables, such as access to health care, trust in the health care system, or societal perceptions of adolescents with ADHD, may confound these associations. Third, the generalizability of this study’s results to populations in regions with different cultural, socioeconomic, and health infrastructure backgrounds is unclear. Whether this study’s results can be generalized to caregivers of adolescents with ADHD who have not sought medical assistance in outpatient settings or who did not participate in the follow-up remains uncertain. Further studies on caregivers of children with ADHD in the community are needed to replicate the results of this study. Finally, 49.5% of the participants were lost to follow-up in this study. There was a risk that the results might be biased due to non-random attrition. Factors influencing whether participants returned for the follow-up could also influence their attitudes towards vaccination, potentially skewing the results. Further follow-up studies employing strategies such as follow-up reminders or incentives may increase the rate of participation at the follow-up assessment. Moreover, the duration between the initial assessment and follow-up varied from 2 to 4 years, which may have affected the reliability of baseline factors as predictors.

## 5. Conclusions

This study discovered that the age of caregivers and their attribution of the etiology of ADHD to disciplinary failures at baseline were predictive of their motivation to vaccinate child against COVID-19 at follow-up. Parenting difficulties during the COVID-19 pandemic were also cross-sectionally associated with this motivation. Based on these findings, this study recommends integrating these factors into the development of intervention programs to enhance the motivation of caregivers to vaccinate their children with ADHD against COVID-19.

## Figures and Tables

**Table 1 vaccines-12-00450-t001:** Caregiver and adolescent characteristics at baseline and follow-up.

	At Baseline			At Follow-Up		
	*n* (%)	Mean (SD)	Range	*n* (%)	Mean (SD)	Range
Caregiver (*n* = 98)						
Gender						
Women	75 (76.5)			75 (76.5)		
Men	23 (23.5)			23 (23.5)		
Age (year)		44.3 (6.0)	30–66		47.7 (6.8)	33–69
Education level						
High school or below	47 (48.0)			47 (48.0)		
College or above	51 (52.0)			51 (52.0)		
Affiliate stigma ^a^		37.9 (10.5)	22–63			
Parenting stress ^b^		0 (1)	−2.6–2.0			
Depression ^c^		9.2 (7.7)	0–28		10.9 (10.1)	0–51
Anxiety ^d^		7.5 (9.3)	0–54		8.0 (9.2)	0–48
Parent’s attribution of the etiology of ADHD						
Denial of brain dysfunction	20 (20.4)			–		
Caregiver’s failure in disciplining	9 (9.2)			–		
Credentialism	28 (28.6)			–		
Parenting difficulties during the COVID-19 pandemic ^e^		–	–		0 (1)	−1.4–3.3
Motivation to vaccinate their child against COVID-19 ^f^		–	–		50.6 (10.1)	15–63
Child (*n* = 98)						
Gender						
Girl	18 (18.4)			18 (18.4)		
Boy	80 (81.6)			80 (81.6)		
Age (year)		12.5 (1.8)	10–16		15.9 (1.8)	13–18
Inattention ^g^		13.2 (6.1)	2–27		10.9 (7.2)	0–27
Hyperactivity/impulsivity ^g^		9.6 (5.7)	0–26		5.7 (5.4)	0–21
Oppositional defiant disorder symptoms ^g^		9.1 (5.9)	0–24		8.9 (6.4)	0–23
Medical adherence		–	–		3.6 (0.7)	1–4

^a^: Measured using the Affiliate Stigma Scale. ^b^: Measured using the Parenting Stress Index. ^c^: Measured using the Beck Depression Inventory-II. ^d^: Measured using the Beck Anxiety Inventory. ^e^: Measured using the Parenting Difficulties in Infectious Disease Pandemic Inventory. ^f^: Measured using the Motors of COVID-19 Vaccination Acceptance Scale. ^g^: Measured using the Parent Form of the Swanson, Nolan, and Pelham Scale, version IV.

**Table 2 vaccines-12-00450-t002:** Prospective and cross-sectional factors related to caregiver motivation to vaccinate children against COVID-19: Bivariable linear regression analysis.

	Prospective Factors at Baseline		Cross-Sectional Factors at Follow-Up	
	B (se)	*p*	B (se)	*p*
Caregiver’s gender ^a^	3.808 (2.573)	0.143	–	–
Caregiver’s age	0.379 (0.192)	0.048	–	–
Caregiver’s education level ^b^	−2.903 (2.192)	0.189	–	–
Caregiver’s affiliate stigma	−0.114 (0.105)	0.282	–	–
Caregiver’s parenting stress	−0.803 (1.098)	0.467	–	–
Caregiver’s depression	−0.113 (0.145)	0.436	−0.093 (0.109)	0.397
Caregiver’s anxiety	−0.143 (0.118)	0.230	−0.147 (0.120)	0.222
Caregiver’s attribution of the etiology of ADHD: denial of brain dysfunction	0.522 (2.706)	0.847	–	–
Caregiver’s attribution of the etiology of ADHD: failure in disciplining	−8.239 (3.677)	0.028	–	–
Caregiver’s attribution of the etiology of ADHD: education systems problems	−2.475 (2.474)	0.320	–	–
Caregiver’s parenting difficulties during the COVID-19 pandemic	–	–	−0.249 (0.106)	0.022
Child’s gender ^c^	2.574 (2.750)	0.352	–	–
Child’s age	0.017 (0.696)	0.981	–	–
Child’s inattention	0.133 (0.181)	0.464	−0.020 (0.155)	0.896
Child’s hyperactivity/impulsivity	−0.105 (0.196)	0.591	0.024 (0.206)	0.909
Child’s ODD	−0.104 (0.187)	0.579	−0.240 (0.172)	0.166
Child’s medical adherence	–	–	2.286 (1.564)	0.148

^a^: Women as the reference. ^b^: High school or below as the reference. ^c^: Girls as the reference. ODD: oppositional defiant disorder.

## Data Availability

The data are available upon reasonable request to the corresponding authors.

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
