# Peer review of "Prospective and Cross-Sectional Factors Predicting Caregiver Motivation to Vaccinate Children with Attention-Deficit/Hyperactivity Disorder against COVID-19: A Follow-Up Study"

_vaccines, 2024, doi:10.3390/vaccines12050450_

Round 1
Reviewer 1 Report
Comments and Suggestions for Authors
Dear authors,
I appreciate your dedicated research efforts; it is both interesting and thoroughly detailed. Your clear delineation of the materials and methods, coupled with a robust statistical analysis, is commendable. The results and discussion sections are also notably well-executed. Furthermore, your acknowledgment of limitations and adherence to proper bibliographic and English language are appreciated.
I would like to suggest, if feasible, the inclusion of the various rating scales used as supplementary material. This would enhance their usability.
Once again, I commend you for your excellent work.
Best regards,
Author Response
Reviewer 1
We appreciated your valuable comments. As discussed below, we have revised our manuscript with underlines based on your suggestions. Please let us know if we need to provide anything else regarding this revision.
Comment
I would like to suggest, if feasible, the inclusion of the various rating scales used as supplementary material. This would enhance their usability.
Response
Thank you for your comment. Regrettably, the six rating scales used in this study are formally developed and copyrighted scales; therefore, we could not included the contents of these scales in our manuscript. Both Motors of COVID-19 Vaccination Acceptance Scale and Parenting Difficulties in Infectious Disease Pandemic Inventory were developed by the corresponding author of this study (CFY) and have included the whole scales in the original articles. The Affiliate Stigma Scale can be used with the consent of the original author (Professor Winnie Wing Sze Mak). The Parenting Stress Index, Fourth Edition Short Form, Beck Depression Inventory-II, and Beck Anxiety Inventory should be purchased from the publishers. The Swanson, Nolan, and Pelham, Version IV Scale has been widely circulated online and can be used freely.
Reviewer 2 Report
Comments and Suggestions for Authors
The study contributes valuable insights into the factors influencing caregiver willingness to vaccinate children with ADHD against COVID-19.
Strengths:
1. The study addresses an important and timely issue – the factors influencing caregiver willingness to vaccinate children with ADHD against COVID-19. Given the ongoing global health challenges posed by COVID-19, understanding these dynamics is crucial for public health planning and interventions.
2. The longitudinal design, with data collected before and during the COVID-19 pandemic, allows for a nuanced understanding of how caregiver attitudes evolve in response to a global health crisis. This prospective approach strengthens the study's ability to identify factors predictive of vaccination willingness.
3. Including a wide range of variables, including caregiver demographics, beliefs about ADHD, parenting stress, and the challenges faced during the pandemic, provides a holistic view of the factors that might influence vaccination decisions.
Drawbacks:
1. The study's participants were recruited from psychiatric outpatient clinics in southern Taiwan, which may limit the generalizability of the findings to broader populations. Taiwan's specific cultural, healthcare and socioeconomic contexts may influence caregiver attitudes in ways that are not fully translatable to other regions or populations.
2. With a follow-up participation rate of 50.5%, there is a risk that the results may be biased due to non-random attrition. Factors influencing whether participants returned for the follow-up could also influence their attitudes towards vaccination, potentially skewing the results.
3. The reliance on self-reported questionnaires introduces the possibility of social desirability bias, where participants might report what they perceive as socially acceptable rather than their true feelings or behaviors. This is particularly relevant for topics like vaccination, which can be polarizing.
Recommendations:
1. Employing strategies to reduce attrition, such as follow-up reminders or incentives for completion, could help ensure a more representative sample at follow-up. Additionally, analyzing potential differences between those who did and did not complete the follow-up could provide insights into the impact of attrition on study findings.
2. Incorporating objective measures of vaccination status or corroborating self-reported data with medical records could strengthen the study's validity. Additionally, employing qualitative methods, such as interviews or focus groups, could provide deeper insights into the motivations behind caregiver attitudes and decisions.
Author Response
Reviewer 2
We appreciated your valuable comments. As discussed below, we have revised our manuscript with underlines based on your suggestions. Please let us know if we need to provide anything else regarding this revision.
Comment
- The study's participants were recruited from psychiatric outpatient clinics in southern Taiwan, which may limit the generalizability of the findings to broader populations. Taiwan's specific cultural, healthcare and socioeconomic contexts may influence caregiver attitudes in ways that are not fully translatable to other regions or populations.
Response
Thank you for your comment. We have listed it as one of the limitations of this study in the original manuscript. Please refer to line 384-389.
“Third, the generalizability of this study’s results to populations in regions with different cultural, socioeconomic, and health infrastructure backgrounds is unclear. Whether this study’s results can be generalized to caregivers of adolescents with ADHD who have not sought medical assistance in outpatient settings or who did not participate in the follow-up remains uncertain. Further studies on caregivers of children with ADHD in community are needed to replicate the results of this study.”
Comment
- With a follow-up participation rate of 50.5%, there is a risk that the results may be biased due to non-random attrition. Factors influencing whether participants returned for the follow-up could also influence their attitudes towards vaccination, potentially skewing the results.
Response
We totally agree with you regarding the possible bias resulted from a high rate of losing follow-up of participants. We added it as one of the limitations of this study as below. Please refer to line 390-395.
“49.5% of the participants lose follow-up in this study. There was a risk that the results might be biased due to non-random attrition. Factors influencing whether participants returned for the follow-up could also influence their attitudes towards vaccination, potentially skewing the results. Further follow-up studies employing strategies such as follow-up reminders or incentives may increase the rate of participation at the follow-up assessment.”
Comment
- The reliance on self-reported questionnaires introduces the possibility of social desirability bias, where participants might report what they perceive as socially acceptable rather than their true feelings or behaviors. This is particularly relevant for topics like vaccination, which can be polarizing.
Response
Thank you for your comment. We have listed it as one of the limitations of this study in the original manuscript. Please refer to line 374-380.
“First, the data collection relied solely on self-reports from caregivers of children with ADHD, potentially introducing single informant bias and social desirability bias. Future studies can enhance validity by incorporating objective measures of vaccination status and corroborating self-reported data with medical records. Additionally, employing qualitative methods, such as interviews or focus groups, could provide deeper insights into the motivations behind caregiver attitudes and decisions.”
Comment
Recommendations:
- Employing strategies to reduce attrition, such as follow-up reminders or incentives for completion, could help ensure a more representative sample at follow-up. Additionally, analyzing potential differences between those who did and did not complete the follow-up could provide insights into the impact of attrition on study findings.
Response
- Thank you for your suggests. We added it as one of the directions for further studies into the revised manuscript. Please refer to line 393-395.
“Further follow-up studies employing strategies such as follow-up reminders or incentives may increase the rate of participation at the follow-up assessment.”
- We have analyzed the differences in demographic characteristics, affiliate stigma, parenting stress, depression, anxiety and beliefs regarding the causes of ADHD between those who did and did not complete the follow-up in Results section of the original manuscript. Please refer to line 267-277.
“The study observed no differences at baseline between those who did and did not complete the follow-up survey in terms of caregivers’ sex (χ2 = 0.783, p = 0.376), age (t = 0.984, p = 0.326), level of education (χ2 = 2.517, p = 0.113), affiliate stigma (t = 0.068, p = 0.945), parenting stress (t = -0.754, p = 0.451), depression (t = 0.437, p = 0.663), anxiety (t = 1.085, p = 0.279), and beliefs regarding the causes of ADHD (χ2 = 0.005~0.687, p = 0.407~0.943). Similarly, no baseline differences were observed in the adolescents’ sex (χ2 = 1.822, p = 0.177), inattention levels (t = 0.074, p = 0.941), hyperactivity/impulsivity (t = -0.525, p = 0.600), and symptoms of ODD (t = 1.126, p = 0.262) between the groups that did and did not participate in the follow-up. However, caregivers who did not participate in the follow-up were more likely to have older children with ADHD (t = 2.807, p = 0.006).”
Comment
- Incorporating objective measures of vaccination status or corroborating self-reported data with medical records could strengthen the study's validity. Additionally, employing qualitative methods, such as interviews or focus groups, could provide deeper insights into the motivations behind caregiver attitudes and decisions.
Response
Thank you for your comment. We incorporated these suggestions into the revised manuscript as below. Please refer to line 376-380.
“Future studies can enhance validity by incorporating objective measures of vaccination status and corroborating self-reported data with medical records. Additionally, employing qualitative methods, such as interviews or focus groups, could provide deeper insights into the motivations behind caregiver attitudes and decisions.”
Reviewer 3 Report
Comments and Suggestions for Authors
This paper reports findings from a study of the prospective and cross-sectional factors influencing caregiver willingness to vaccinate children with ADHD against COVID-19. The study follows standard protocols for this category of studies and the paper is generally well written.
The main recommendation for revision of the paper that I would make is that, given the limited sample size and the single population context of the study, other studies with different samples and social/cultural contexts should be done in order to assess the generalizability of the findings.
Author Response
Reviewer 3
We appreciated your valuable comments. As discussed below, we have revised our manuscript with underlines based on your suggestions. Please let us know if we need to provide anything else regarding this revision.
Comment
The main recommendation for revision of the paper that I would make is that, given the limited sample size and the single population context of the study, other studies with different samples and social/cultural contexts should be done in order to assess the generalizability of the findings.
Response
Thank you for your comment. Please refer to line 386-395.
“Whether this study’s results can be generalized to caregivers of adolescents with ADHD who have not sought medical assistance in outpatient settings or who did not participate in the follow-up remains uncertain. Further studies on caregivers of children with ADHD in community are needed to replicate the results of this study.”
“49.5% of the participants lose follow-up in this study. There was a risk that the results might be biased due to non-random attrition. Factors influencing whether participants returned for the follow-up could also influence their attitudes towards vaccination, potentially skewing the results. Further follow-up studies employing strategies such as follow-up reminders or incentives may increase the rate of participation at the follow-up assessment.”
Reviewer 4 Report
Comments and Suggestions for Authors
In the intro: "The pandemic had 42 particularly adverse effects on children and adolescents ". It could be argued that many other groups, especially elderly or immunocompromised, who were dying are much more adversely affected
In methods, can you specify how you got 98 from 194? How many were excluded? How many simply did not follow up? I do not follow the sex of the caregiver in Table 2. What if the child lived with both parents? In the interest of transparency and replicability, the actual dataset (anonymized) should be made availableAuthor Response
Reviewer 4
We appreciated your valuable comments. As discussed below, we have revised our manuscript with underlines based on your suggestions. Please let us know if we need to provide anything else regarding this revision.
Comment
In the intro: "The pandemic had particularly adverse effects on children and adolescents". It could be argued that many other groups, especially elderly or immunocompromised, who were dying are much more adversely affected
Response
Thank you for your comment. We revised this sentence and added descriptions regarding the high rates of morbidity and morbidity in the elders and people with chronic diseases and immunocompromise. Please refer to line 42-50.
“Older individuals have been identified as the population with high mortality and morbidity rates of COVID-19 [3]. In addition to older individuals, the individuals with chronic diseases such as hypertension, chronic obstructive pulmonary disease, obesity, cerebrovascular disease, chronic liver disease, chronic renal disease, human immunodeficiency virus infection, tuberculosis and cancer have a greater probability of mortality and morbidity after contracting COVID-19 [4–6]. Review studies have concluded that despite the lower infection rate in children compared to adults, children remain susceptible to COVID-19 [7,8]; those with comorbidities are especially susceptible [9].”
Comment
In methods, can you specify how you got 98 from 194? How many were excluded? How many simply did not follow up? I do not follow the sex of the caregiver in Table 2. What if the child lived with both parents? In the interest of transparency and replicability, the actual dataset (anonymized) should be made available.
Response
- Thank you for your comment. In this study, 98 caregivers participated in the follow-up study, 10 refused participation, and 86 stopped visiting the outpatient clinics and lose follow-up. We added the explanation into Results section. Please refer to line 265-267.
“In total, 98 participants (75 women and 23 men), representing 50.5% of the initial group, participated in the follow-up study, 10 (5.2%) refused participation, and 86 (44.3%) stopped visiting the outpatient clinics and lose follow-up.”
- This study included caregivers who were the medical decision makers for this adolescent, whether they were a father or mother. We added the explanation into Methods section. Please refer to line 147.
“Individuals were included if they were … the medical decision makers for this adolescent.”
- The data are available upon reasonable request to the corresponding authors. We stated it in Data Availability Statement. Please refer to line 419-420.